# Medicines Optimisation for Respiratory Patients: The Establishment of a New Consultant Respiratory Pharmacist Role in Northern Ireland

**DOI:** 10.3390/pharmacy9040177

**Published:** 2021-10-30

**Authors:** Cairine Gormley, Maureen Spargo, Glenda Fleming, Brendan Moore, Michael Scott, Rose Sharkey, Anne Friel

**Affiliations:** 1Pharmacy Department, Altnagelvin Hospital, Western Health and Social Care Trust, Derry BT47 6SB, UK; Brendan.Moore@westerntrust.hscni.net (B.M.); Anne.Friel@westerntrust.hscni.net (A.F.); 2Medicines Optimisation and Innovation Centre, Antrim BT41 2RL, UK; Maureen.Spargo@northerntrust.hscni.net (M.S.); Glenda.Fleming@northerntrust.hscni.net (G.F.); DrMichael.Scott@northerntrust.hscni.net (M.S.); 3Respiratory Department, Altnagelvin Hospital, Western Health and Social Care Trust, Derry BT47 6SB, UK; Rose.Sharkey@westerntrust.hscni.net

**Keywords:** consultant, respiratory, pharmacist, optimisation, pharmacist services, service development and implementation, consultant pharmacist, respiratory pharmacy, medicines optimisation

## Abstract

Medicines optimisation for those with respiratory conditions can have a significant impact on clinical outcomes and substantial efficiency gains for health care. Consultant pharmacists are experts working at the top of their specialism in four main pillars of practice, namely clinical care, leadership, education and training, and research and development. A consultant respiratory pharmacist has recently been appointed at a large Health and Social Care Trust in Northern Ireland to provide expert care and clinical leadership for the medicines optimisation agenda with regards to respiratory care in Northern Ireland. Alongside clinical practice, leadership, and service development, emphasis will be placed on monitoring and evaluating the work of the consultant respiratory pharmacist with a view to gathering the necessary evidence to support the case for further investment in such consultant pharmacist posts in the region. This short communication article outlines some of the clinical and economic factors associated with the decisions to invest in the consultant pharmacist model of care in Northern Ireland

## 1. Introduction

Respiratory disease is a major cause of morbidity and mortality. It is the third most common cause of death in Northern Ireland (NI), accounting for 13.8% of deaths in 2018 [1]. Caring for people with chronic respiratory disease costs the United Kingdom (UK) £11 billion per year and places substantial demand on health and social care resources [2]. Following the COVID-19 global pandemic, there will be an increase in respiratory disease in those exposed to the virus.

The Northern Ireland Medicines Optimisation Quality Framework aims to improve the health outcomes of the NI population by supporting best practice in the prescription, supply, and administration of medicines [3]. Furthermore, in response to the WHO Global Patient Safety Challenge to reduce avoidable medicines-related harm by 50% over the next five years, NI health officials have developed an action plan that aims to support and empower people to become more involved in decisions regarding their medicines, reduce inefficiencies and errors at transitions of care, and improve adherence to treatments [4].

The treatment of respiratory disease is often complex and those living with chronic respiratory conditions are often prescribed many medications with complex regimens. The aging population has major implications for the provision of respiratory services to certain cohorts, such as older people, who often have complex social and emotional support needs in addition to high rates of polypharmacy (the prescription of multiple medications), in the context of multiple long-term conditions.

Consultant pharmacist roles were first established in the UK in 2005 and those who are working at this level are regarded as clinical leaders in their field and profession [5]. A consultant pharmacist is a pharmacist who has developed and demonstrated high-level expertise in their area of practice and across four pillars, namely clinical practice, leadership, education, and research [5]. Depending on resources, a consultant pharmacist will often lead a team of clinical pharmacists. The consultant pharmacist provides care for more complex patients, and clinical pharmacists can refer patients to the consultant pharmacist for advice. The consultant pharmacist liaises with all professionals in the various pillars of practice. They provide expert care to patients with the most complex needs, such as palliative care and those with chronic respiratory disorders. Despite the framework for consultant pharmacist posts being in place for 15 years, just six pharmacists are currently working at this level in NI: five within the care of older people and one within oncology.

In this communication paper, the process for establishing a consultant pharmacy-led respiratory service at the Western Health and Social Care Trust (WHSCT) in Northern Ireland and the plans for development are discussed.

## 2. Clinical and Economic Factors

The need for a consultant respiratory pharmacist for the WHSCT became apparent following a series of service development projects carried out at the Trust [6,7,8]. In 2013, deaths due to respiratory disease within the WHSCT area accounted for 16.2% of all deaths at the trust (2% higher than the NI average). This prompted innovative medicines optimisation service development work that focused on case management of patients with chronic obstructive respiratory disease (COPD) by a specialised hospital pharmacist in both primary and acute care settings. One pilot [6,7,8] brought specialist pharmacist case management medicines optimisation clinics to COPD patients in a selection of GP-based clinics between 2015 and 2016. During these clinics, the pharmacist first determined a patient’s disease stage using the GOLD grading system [9]. They then established COPD medication appropriateness and medication adherence. All clinical interventions made were recorded and graded for significance. The pharmacist also prescribed COPD medications and smoking cessation support, determined whether antibiotic prescribing for exacerbations followed local and national guidelines, and made appropriate referrals to other healthcare professionals in both primary and secondary care (respiratory specialist review, spirometry, long-term oxygen therapy assessment, pulmonary rehabilitation, smoking cessation). Patients were followed up for 30 days post-baseline clinic.

Results for 646 patients seen over a six-month period demonstrated significant improvements (Wilcoxon Signed Rank Test, *p* < 0.001) in COPD medication appropriateness, improvements in COPD symptoms, and improved guideline-informed antibiotic prescribing. Annual drug cost savings were £244k [6,7,8]. A sub-cohort (*n* = 360) of the patients who attended the clinics were followed up after 12 months. Over two thirds (68%) of the patients had experienced one or more COPD exacerbations during the year prior to clinic attendance. This reduced to 50% during the 12 months post-intervention. Non-elective COPD-related hospital admissions also decreased (9.2% versus 5.3% over 12 months) [8].

The case management clinics were later introduced to more GP surgeries across the Trust to test for reproducibility. Clinics were established across several GP practices and involved 240 patients. The pharmacist made an average of three clinical interventions per patient (range, 1–8); all interventions indicated a significant improvement in the standard of patient care. At follow-up, the main outcomes were improved appropriateness of antibiotic prescribing at the 30-day review, improved adherence to COPD medicines and appropriateness of COPD prescribing, and improved COPD symptoms. The average net drug cost saving was £43.54 per patient per annum [10].

In September 2016, a case management medicines optimisation service for COPD was implemented for patients admitted to an acute care hospital within the Trust [11]. The aim of this pilot project was to evaluate a specialist pharmacist-led service for COPD patients admitted to the hospital who may or may not be under the care of a respiratory consultant. Referrals were made by the respiratory consultants, clinical pharmacists, and nurses or identified through routine review of new admissions. After confirming medical history and medications taken, the review focused on COPD symptoms, medicines optimisation, and adherence together with education and referral to other services. Patients were followed up, with consent, via telephone approximately 30 days later.

The pharmacist reviewed 107 patients, 69 (64.5%) of whom were experiencing a COPD-related admission. Forty-four patients (41.1%) were directly under the care of a respiratory consultant. The pharmacist made 4.1 ± 2.3 clinical interventions per patient. The main outcomes were improved appropriateness of prescribing and adherence to COPD medicines, and improvement in symptoms between baseline and 30-day review. An average drug cost saving of £86.64 per patient per annum was achieved. The number and frequency of exacerbations were also reduced after the intervention. This work was recognised in the 2016 Health Service Journal Awards (HSJ), where the service development won the top award for medicines optimisation. The findings also indicated that a consultant pharmacist for respiratory medicine could provide the clinical leadership needed to drive forward these innovations in the management of people with respiratory disease.

In addition to the management of COPD, expert pharmacist input could also be of benefit in the management of asthma. In a recent report of all items prescribed from general practice between October and December 2020 [12], seven inhalers featured in the top 40 most costly drugs prescribed in the NI. The total cost to the health service was in excess of £3.5 million. The report also revealed that 7.1% of patients had received more than 12 prescriptions for a short-acting beta agonist inhaler in 2020. The National Review of Asthma Deaths published in 2014 [13] recommended that all asthma patients who have been prescribed more than 12 short-acting reliever inhalers in the previous 12 months should have an urgent review of their asthma control. A consultant-led respiratory pharmacy service, consisting of a consultant pharmacist, advanced practice pharmacists, and pharmacy technicians, could provide the clinical input required to achieve standardised medicines optimisation for respiratory care at the Trust and across NI.

## 3. Establishment of the Consultant Pharmacist Role

In NI, approval for the establishment of consultant pharmacist posts is provided by the Department of Health [5]. Input into the planning of the consultant pharmacist role was sought from representatives from senior management in pharmacy and respiratory care, senior medical consultants in respiratory care, and representatives from the Department of Health and a postgraduate education and learning organisation. A consultant pharmacist post application pack—consisting of the intended job plan for the role, its underpinning structures, details of case load referrals and management, and a full job description and personal specification—was compiled.

Funding was awarded, which included funding from the Medicines Optimisation and Innovation Centre (MOIC), to provide support for research and service development activity. MOIC is a regional centre in NI dedicated to delivering medicines optimisation to the people of NI.

Updated guidance on consultant pharmacist posts [5] indicates that roles should span four key pillars of practice: clinical practice, leadership, education, and research. These pillars of practice correlate strongly with the Advanced Pharmacy Framework [14]. Using these guidance documents, a nominal job plan for the consultant respiratory pharmacist post at WHSCT was proposed to the Department of Health (Table 1) and subsequently approved.

Accreditation of an individual to be considered for a consultant pharmacist role is a separate process. There is no specific postgraduate university qualification to become a consultant pharmacist. However, in line with postgraduate development of all clinical pharmacists, all post-holders would be expected to have completed a diploma in clinical pharmacy and be a pharmacist-independent prescriber. The current post-holder of the consultant respiratory pharmacist role at WHSCT was required to demonstrate their competence and expertise to a multidisciplinary and multi-agency interviewing panel. The process of credentialing of consultant pharmacists is currently under review in NI. It is expected that the process will closely follow the system used by the Royal Pharmaceutical Society (RPS) in Great Britain. The RPS has developed an individual credentialing process for advanced-level pharmacists. Pharmacists are assessed through this process and, if successful, added to a register of credentialed individuals and will then be able to apply for consultant pharmacist posts.

## 4. Early Evidence of Impact

The consultant respiratory pharmacist was appointed at WHSCT shortly before the second wave of COVID-19 pandemic, and was immediately deployed to the COVID-19 Respiratory Support Unit (RSU) to provide an advanced pharmaceutical care and prescribing service to patients admitted to the unit. Medicines optimisation in patients with COVID-19 presented new challenges for healthcare staff. Patients were often too ill to assist with medicines reconciliation and relatives were not permitted to be present at the bedside. Clinical challenges included the changing anticoagulation advice as new evidence emerged, anticoagulation of patients who were too ill to weigh, and the management of steroid-induced hyperglycaemia. Rationalisation of medication administration times to decrease nurse exposure during medicines administration also required careful consideration. The input of a consultant pharmacist in the COVID RSU was particularly important as they were able to apply a high level of expertise and make decisions in the absence of evidence or when there was conflicting evidence, as was the case in the management of patients with COVID-19. This application of knowledge and experience is above and beyond the competence level of a clinical pharmacist.

The consultant respiratory pharmacist collected data on all interventions made during deployment at the RSU. Over a 10-week period, 889 interventions were made, including 120 medicines reconciliations and 247 medications were prescribed by the consultant pharmacist. Some 140 medicines were stopped or had a stop date added to the prescription. The clinical significance of each intervention was self-graded using the Eadon Criteria [15]: a scale from 1 to 6, where a score of 4 or greater represents an improvement in the quality of patient care. The majority of interventions (*n* = 884; 99.4%) were graded 4 or greater. A proportion of interventions (10% of grade 4 and all grade 5 and grade 6) were independently reviewed by a consultant respiratory physician. Furthermore, the University of Sheffield School of Health and Related Research (ScHARR) model [16] was applied to estimate potential cost savings associated with the interventions made by the consultant respiratory pharmacist. Findings (based on the consultant respiratory physician scores) indicated potential savings of between £99,000 and £192,000 [17]. The ScHARR model “significant adverse drug event” intervention (Eadon grade 4) cost range per intervention was £65–150, “serious adverse drug event” intervention (Eadon grade 5) cost range per intervention was £810–1232, and “Severe, life-threatening or fatal adverse drug event” intervention (Eadon score 6) cost range per intervention was £1232–1760. The results are displayed in Table 2.

## 5. Future Plans for the Consultant Pharmacist Service

The consultant respiratory pharmacist continues to act as a leader within the pharmacy team for the care of patients with COVID-19 and other respiratory conditions. The consultant acts in an advisory role within the multi-professional team, providing advice and treatment recommendations where there is limited evidence for the condition. When the third wave of COVID-19 receded, the consultant respiratory pharmacist was able to focus again on enacting the agreed job plan for the role.

Fifty percent of the consultant respiratory pharmacist role is patient facing. This includes attending the daily post-take ward rounds on the respiratory ward and respiratory outpatient clinics to provide specialist advice. The post-holder will continue to provide expertise to the care of the asthmatic patient and is developing a difficult asthma outpatient clinic to optimise medication and improve adherence for this patient group. The aim of this service will be to improve patients’ symptom control and quality of life and reduce frequent readmissions to hospital. This clinic will expand to include other respiratory conditions, such as bronchiectasis and COPD.

Outcome measures that will be used to measure the impact of the consultant pharmacist on the asthma patient will be similar to COPD outcomes listed above and will also include:Percentage of patients with peak flow measurements completed and peak flow diary supplied.Percentage with inhaler technique assessed.Medicines optimisation review with Eadon classification of interventions and ScHARR drug expenditure recording.Percentage with Personal Asthma Action Plan provided.Asthma Control Test Questionnaire results recorded at follow-up appointments.

The consultant respiratory pharmacist also plans to standardise care for respiratory patients across the region by leading on the development of respiratory medicines optimisation education and training for GPs, nurses, pharmacists, and pharmacy technicians working across primary and secondary care. A regional group for pharmacists with an interest in respiratory medicine will be established, which will work collaboratively to develop training materials, guidelines, procedures, and protocols, and a local respiratory medicine formulary. Links with similar interest groups in the rest of the UK will be made to keep abreast of new developments and shared learnings opportunities.

Support for research and service development activity is being provided by MOIC. This novel collaboration of clinical and research leadership will help to support the growth of the post-holder’s research and development portfolio, and increase the capacity of the pharmacy workforce to conduct high-quality research. In addition, it will generate and disseminate research into medicines optimisation and care of the respiratory patient in NI. Projects that are in the pipeline include implementing innovative ways for patients to take ownership of their condition and its management, e.g., the use of mobile apps to enhance adherence and inhaler technique, and the development of electronic shared care action plans between primary and secondary care. The consultant respiratory pharmacist will also strive to gather evidence of the economic impact that the consultant pharmacist model could bring to the region and help build the business case for the funding of similar advanced practice and consultant pharmacist posts.

## 6. Conclusions

Consultant pharmacist posts in the UK provide opportunities for clinical pharmacists to demonstrate and build upon their clinical leadership skills and preside over a wider sphere of influence within the healthcare system. The appointment of the first consultant pharmacist for respiratory care in NI has been a welcome development for the region and has been driven by a clearly identified need to improve medicines use in people living with respiratory disease there. There is an imperative now for the post-holder to generate and disseminate robust evidence of their impact on patient care and return of investment.

## Figures and Tables

**Table 1 pharmacy-09-00177-t001:** Job plan details and allocation of time for the consultant respiratory pharmacist role at the Western Health and Social Care Trust.

Pillars of Practice	Corresponding Advanced Pharmacy Framework Cluster	% Job
Clinical practice	Expert professional practice Expert skills and practice Delivery of professional expertise Reasoning and Judgement Professional Autonomy	50%
Leadership	Leadership & Management Strategic context Governance Vision Innovation Service development Motivational	10%
Education	Education Training and Development Role Model Mentorship Conducting education and training Professional Development Links Practice with Education Educational Policy	20%
Research	Research and Evaluation Critical Evaluation Identifies gaps in evidence base Develops and Evaluates Research Protocols Creates Evidence Research Evidence into Working practice Supervises others undertaking research Establish research partnership	20%

**Table 2 pharmacy-09-00177-t002:** Estimation of potential cost savings associated with interventions.

Eadon Criteria (Grade)	Number of Interventions	ScHARR Lower Limit (£) Estimation	ScHARR Upper Limit (£) Estimation
Significant: improvement in the standard of care (Grade 4)	826	53,690	123,900
Very significant: prevents major organ failure or adverse reaction of similar importance (Grade 5)	53	42,930	65,296
Potentially life-saving (Grade 6)	2	2464	3520
Totals	881	99,084	192,716

## Data Availability

No new data were created or analysed in this study. Data sharing is not applicable to this article.

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
