# Peer review of "Medicines Optimisation for Respiratory Patients: The Establishment of a New Consultant Respiratory Pharmacist Role in Northern Ireland"

_pharmacy, 2021, doi:10.3390/pharmacy9040177_

Round 1
Reviewer 1 Report
This paper describes advancement of clinical pharmacists' roles in an ever-increasingly complex healthcare system where their input can make a significant impact on the outcomes of respiratory (and other) patients. It is important that the healthcare system learns about the development and impact of these roles. Section 2 could be more explicit describing the interface settings where the medicines optimisation service for COPD patients had shown benefit. This section is important as it describes the reason why the CP role was set up in Northern Ireland- excess deaths, use of costly medicines and inappropriate (over) use of respiratory medicines - but could these be more explicitly stated given they would presumably be some of the measures used to assess impact of the CP role in the future. Section 4 of the commentary could be improved by describing how the consultant pharmacists role in the COVID-19 Respiratory Support Unit differs from the role a clinical pharmacist could make in this Unit. (It seems the establishment of respiratory CP role as first designed was interrupted and diverted by advent of COVID-19. It would be good to be more explicit how the CP role made a difference to usual care albeit there was no trial). Also, a table of the potential savings from the interventions may be easier for the reader to digest. Please describe who undertook the grading of the interventions. Section 5 could be improved by explicitly stating the outcome measures that will be used to measure the impact of the CP (maybe in a Box?) as these will be important information for those thinking of developing a similar model of care. It was also unclear from the commentary whether the CP had a role in pulmonary rehabilitation and whether they were liaising with the medical profession in the various pillars of practice. The dissemination of the business case for CP role is also very important as this is frequently a stumbling block for the implementation of advanced roles.
Author Response
"Please see the attachment"

Reviewer 2 Report
Thank you for your submission. It is quite interesting. It would help readers, especially from other parts of the world, if the term “consultant pharmacist” was explained more. How does one achieve that status? Does it require postgraduate university qualifications? Is there an assessment? Reference 5 is cited, but it would be appropriate to include some more details in the manuscript. Were there respiratory clinical pharmacists prior to this position being established (which would be routine in many hospitals?) Does this position replace a clinical pharmacist? How does this position work with hospital clinical pharmacists? Do they also work with GPs and community pharmacists?
As noted, “There is an imperative now for the post-holder to generate and disseminate robust evidence of their impact on patient care and return of investment.” Along those lines, how exactly was the clinical significance of their interventions assessed? (“The clinical significance of each intervention was graded using the Eadon Criteria [13] — a scale from 1 to 6 where a score of 4 or greater represents an improvement in quality of patient care.”). Was this performed by an independent panel?
Author Response
"Please see the attachment"
